# Latitudinal Difference in the Condition Factor of Two Loliginidae Squid (Beka Squid and Indian Squid) in China Seas

Jianzhong Guo [1,2,3], Chi Zhang [1], Zhixin Li [1], Dan Liu [1] and Yongjun Tian [1,2,*]

1   Research Centre for Deep Sea and Polar Fisheries, and Key Laboratory of Mariculture, Ministry of Education, Ocean University of China, Qingdao 266100, China; gjz@qau.edu.cn (J.G.); zhangchi6962@ouc.edu.cn (C.Z.); 21190531176@stu.ouc.edu.cn (Z.L.); 21210511043@stu.ouc.edu.cn (D.L.)
2   Frontiers Science Center for Deep Ocean Multispheres and Earth System (FDOMES), Ocean University of China, Qingdao 266100, China
3   School of Marine Science and Engineering, Qingdao Agricultural University, Qingdao 266109, China
*   Correspondence: yjtian@ouc.edu.cn

**Abstract:** Cephalopod fisheries in the China Seas have witnessed an increasing trend in the catches of coastal cephalopods since the 1990s, with Loliginidae squid emerging as the main commercial target species. However, climate change and overfishing have led to a dramatic reduction in Loliginidae squid resources, highlighting the need to improve monitoring, protection, and management of this species. The *Loligo beka* and *Uroteuthis duvaucelii* are widely distributed along the coastal areas of the China Seas, and have commercial and ecological importance. Despite having overlapping distributions, similar life histories, and a strong dependence on the marine environment, there is limited knowledge about their growth and responses to environmental changes, hindering the effective management of their resources. In this study, we investigated the interspecies and intra-species differences in condition factor and their responses to temperature changes by analyzing data collected from wide coastal areas of the China Seas from June 2019 to November 2020. The findings showed that both species exhibited allometric growth and reproduced throughout the year, with two main breeding peaks. There were significant monthly variations and latitude differences in the intra-species growth, with a higher proportion of small-sized individuals (between 5 and 10 g for *L. beka* and between 10 and 20 g for *U. duvaucelii*) in low-latitude waters. The latitudinal differences in body weight and distribution between and within the two species were mainly due to natural habitats, especially temperature. Our mixed effect model results demonstrated that both species' body weight increased with increasing temperature, suggesting that Loliginidae squid have significant environmental adaptability and can be used as an indicator species for studying environmental changes in the China Seas. These findings have significant implications for understanding the population dynamics, species development, and regionally specific management of Loliginidae squid fisheries in the China Seas.

**Keywords:** *Loligo beka*; *Uroteuthis duvaucelii*; Loliginidae squid; body weight; condition factor; China Seas



## 1. Introduction

Cephalopods play a critical role in marine ecosystems as they occupy various trophic levels and have become an important commercial resource in the world [1,2]. The increase in cephalopod catches has maintained the global marine fisheries at a relatively stable and high level [3]. Thus, cephalopods have become the driving force of ecosystem changes and indicators of potential climate change [4]. Due to overfishing of finfish competitors and prey, cephalopods have experienced a reduction in competitive pressure, leading to their increased importance in marine ecosystems [5,6]. Moreover, cephalopods are ecologically and commercially important invertebrates that are widely distributed in all marine habitats from the polar to tropical waters, except for the hadal zone and the Black Sea [7]. Their

unique life history characteristics, including short lifespan, high natural mortality and turnover rate, rapid and non-asymptotic growth, and diversified habitats, make them capable of quickly adapting to environmental changes, giving them a competitive advantage over other biological populations and complex population structures [2,8,9].

The China Seas are an overexploited coastal ecosystem and one of the seas with drastic environmental changes, which are gradually warming up with climate change [10]. In the China Seas, cephalopod fisheries have increased since the 1990s, while commercial fish stocks have largely declined, and individual fish tend to be small-sized due to the drastic environmental changes caused by climate change [10,11]. Chinese coastal cephalopods are characterized by a wide range of species, significant latitude differences, complex population structure, and the population size, distribution, and fluctuations are greatly affected by the marine environment and climate change [11,12]. However, there is a lack of long-term survey data and studies on the dynamics of Chinese coastal cephalopods [11,13], specifically Loliginidae squids, which severely affect the reliability of population assessment and the effectiveness of fishery management of cephalopods in China. Loliginidae squids are an important part of the coastal ecosystem and the main commercial target species of cephalopod fisheries [14], making them ideal for studying population dynamics and response to climate change [15,16]. Eight Loliginidae squid species distribute in China's coastal waters, of which five species (*Uroteuthis* (*Photololigo*) *duvaucelii*, *U. edulis*, *U. chinensis*, *Loligo beka*, and *L. japonica*) are commercially exploited and mainly inhabit the tropical and temperate waters [17,18]. Furthermore, their population structure is complex, with relatively few previous studies and a lack of data, which limits our understanding of population dynamics and hinders the management and development of resources.

*Loligo beka* and *Uroteuthis duvaucelii* are shallow-sea species of Loliginidae squid with a complex population structure, short lifespan, high growth plasticity, rapid generation change, and are also commercially important species in cephalopod fisheries in China [19,20]. *L. beka* is most commonly a small-sized squid, mainly distributed in the temperate to tropical waters of the Western Pacific, and widely distributed in the coastal waters of China and southern Japan [21]. *U. duvaucelii* is a medium-sized squid, widely distributed in the Pacific Ocean and the Indian Ocean, especially in the Red Sea, Arabian Sea, and coastal waters of India [22]; additionally, it is widely distributed in the East China Sea and South China Sea [20,23]. The squids have the habit of diel migration, swimming to the surface at night, and are caught as bycatch in set and trawl nets [24]; they mainly feed on benthic species of crustaceans, fishes, and squids [21]. They have two spawning groups, namely, the spring spawning cohort (SSC) and autumn spawning cohort (ASC), and the fishing season is mainly carried out according to their spawning and overwintering groups. For example, the fishery in the Yellow Sea is mainly carried out in spring to target spawning groups and catch overwintering groups in winter [24]. Growth status of the squids are all highly sensitive to environmental change and have potential value in monitoring environmental change [25]. Studies on Loliginidae squid in China have mainly focused on *U. edulis* and *U. chinensis* [26–30], while *L. beka* and *U. duvaucelii* have received little attention [31,32]. Especially, the growth status of the China Sea is not clear in a large range across latitudes, which is not good for resource management.

Conditional factors are essential biological parameters and important indicators for assessing the status of aquatic species, populations, and communities [33,34]. Among them, growth is very important for estimating life history, population structure, demographics, ecosystem dynamics, and fishery sustainability [35,36]. The growth characteristics of cephalopods can help identify their population structure and improve our understanding of their key biological processes [37,38]. The spatial difference of squid growth mainly depends on body size and environmental conditions—especially temperature—of individuals. Furthermore, temperature is the most important factor affecting the growth rate and lifespan of species [39]. Perez et al. [15] showed that squid growth has great plasticity with temperature changes over its wide geographical range.

In this study, we sampled the coastal waters of China from the north to the south, and analyzed the latitudinal difference in the condition factor of *Loligo beka* and *Uroteuthis duvaucelii* and the effect of temperature on its body weight by using the mixed effect model. The aim was to: (1) provide a better understand on weight–growth pattern of *L. beka* and *U. duvaucelii*; and (2) understand the response of Loliginidae squid species to environmental warming in order to facilitate future management and development of cephalopod resources in the China Seas.

## 2. Materials and Methods

### 2.1. Study Area and Data Sources

From June 2019 to November 2020, monthly sampling was conducted off the coast of the China Seas, with the exception of January–February 2020 due to outbreak concerns. The study area encompassed the Northern Yellow Sea (NYS), Central Yellow Sea (CYS), East China Sea (ECS), Southern Yellow Sea (SYS), and South China Sea (SCS) (Figure 1; Table 1). The sample collection procedure involved capturing *L. beka* and *U. duvaucelii* specimens, with a larger number of *L. beka* specimens (933, 868, and 676 from NYS, CYS, and ECS, respectively) obtained from the Northern Yellow Sea, Central Yellow Sea, and East China Sea. Meanwhile, only 60 *L. beka* specimens were captured from the Southern Yellow Sea, and 2 were captured from the South China Sea. *U. duvaucelii* specimens (1364, 1251, and 1036) were mainly distributed in the East China Sea, Northern South China Sea, and Beibu Gulf, respectively. The samples were randomly selected from survey samples and local commercial landings and collected through bottom trawl, light luring, or squid jigging. The samples were stored in ice and transported to the laboratory for subsequent biological analysis. The dorsal mantle length (ML, mm) and body weight (BW, 0.1 g) of each individual were measured, and the gonadal development stage was visually assessed based on the Lipinski and Underhill method [40]. The study selected *L. beka* samples from the Northern Yellow Sea, Central Yellow Sea, and East China Sea, and *U. duvaucelii* samples from the East China Sea, Northern South China Sea, and Beibu Gulf based on the collected information to investigate the latitudinal difference in their condition factor.

### 2.2. Marine Environmental Data

As a marine environmental variable, sea surface temperatures (SST) are believed to play a crucial role in determining the optimal habitat for cephalopods, as well as affecting their population structure, size, and distribution [3]. To investigate the influence of temperature on the body weight of two cephalopod species, monthly average SST data for each region were obtained from the Moderate-Resolution Imaging Spectroradiometer (MODIS) Aqua products of the National Oceanic and Atmospheric Administration (NOAA) website (https://coastwatch.pfeg.noaa.gov/, accessed on 10 May 2021). The spatial resolution of all marine environmental data is 0.04166°.

### 2.3. Statistical Analysis and Model Building

The statistical analysis and model building process were performed using body weight as an indicator to evaluate the condition factor of the two species, and regression and correlation analysis were carried out on mantle length and body weight. The multivariate analysis of variance (MANOVA) was employed to compare the differences in body weight between months and regions, while the length–weight relationship (LWR) was determined using the following equation [33]:

$$BW = aML^b$$

where $a$ and $b$ are the coefficient and exponent of the arithmetic form of length–weight relationship, respectively. The value of parameter $b$ reflected the isometric ($b = 3$) or allometric ($b \neq 3$) growth pattern of species [41].

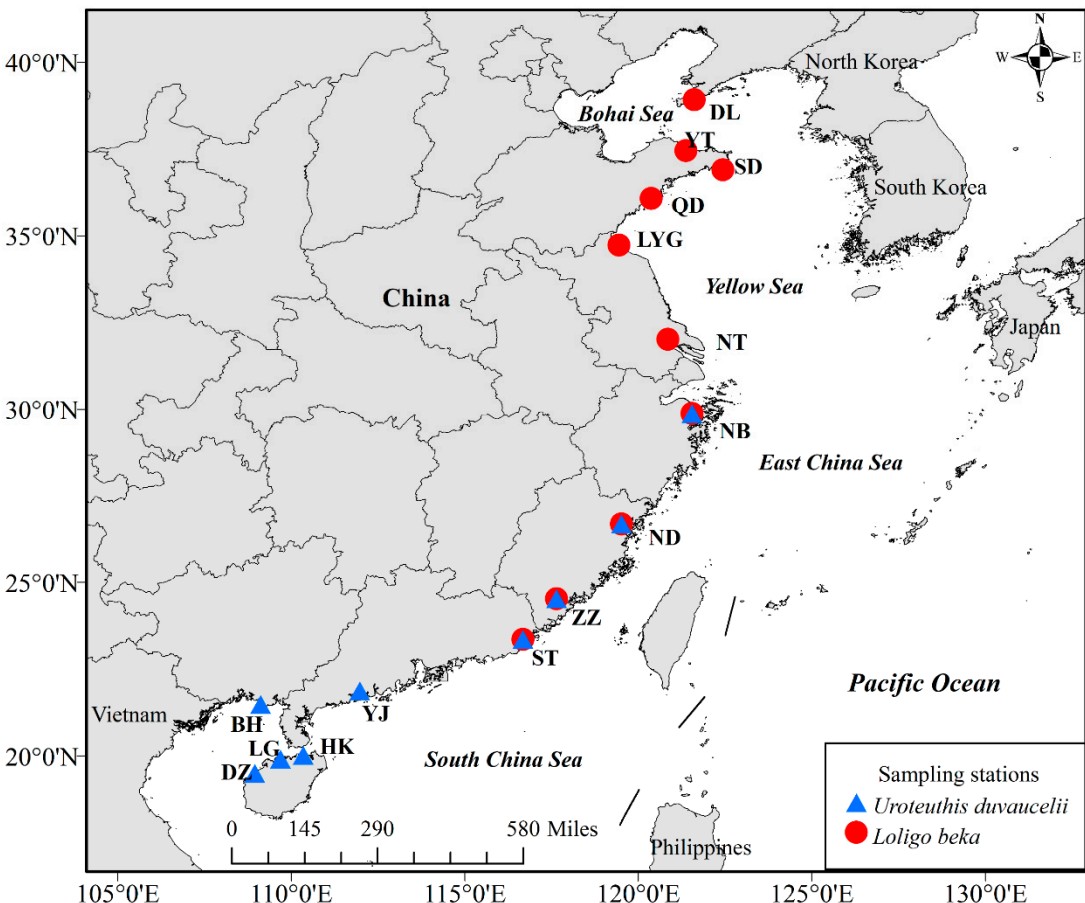

**Figure 1.** Sampling stations of the *Loligo beka* and *Uroteuthis duvaucelii* in China Seas (DL: Dalian; YT: Yantai; SD: Shidao; QD: Qingdao; LYG: Lianyungang; NT: Nantong; NB: Ningbo; ND: Ningde; ZZ: Zhangzhou; ST: Shantou; YJ: Yangjiang; BH: Beihai; HK: Haikou; LG: Lingao; DZ: Danzhou).

**Table 1.** Sample data for *Loligo beka* and *Uroteuthis duvaucelii* in the China Seas.

| Species | Region | Stations | Numbers | Year | Month | Body Weight/g | | Mantle Length/mm | |
|---|---|---|---|---|---|---|---|---|---|
| | | | | | | **Range** | **Mean** | **Range** | **Mean** |
| *Loligo beka* | NYS | DL | 660 | 2019 2020 | 9–12 1, 4–9 | 0.5–20.6 | 6.6 ± 2.9 | 14–78 | 44.6 ± 8.0 |
| | | YT | 103 | 2019 | 6, 7 | 2.4–40.8 | 13.5 ± 8.1 | 42–83 | 61.2 ± 10.7 |
| | | SD | 170 | 2019 | 9, 10, 12 | 1.2–18.4 | 5.6 ± 3.9 | 20–65 | 39.9 ± 9.4 |
| | CYS | QD | 659 | 2019 2020 | 6–12 1, 4, 8, 9 | 0.8–41.3 | 8.6 ± 5.3 | 19–87 | 47.1 ± 12.5 |
| | | LYG | 209 | 2019 2020 | 6, 7 7 | 2.6–24.3 | 9.8 ± 4.2 | 32–72 | 51.1 ± 7.2 |
| | SYS | NT | 60 | 2020 | 5 | 8.7–43.1 | 23.7 ± 8.8 | 45–83 | 61.4 ± 9.0 |
| | ECS | NB | 240 | 2019 2020 | 8–10 4 | 1.4–29.1 | 7.6 ± 3.8 | 22–68 | 45.8 ± 8.7 |
| | | ND | 386 | 2019 2020 | 12 1, 5–9 | 2.4–31.3 | 9.2 ± 3.8 | 27–83 | 47.6 ± 7.2 |
| | | ZZ | 50 | 2019 | 8 | 3.5–12.2 | 7.3 ± 2.2 | 38–58 | 46.6 ± 4.5 |
| | NSCS | ST | 2 | 2019 | 8 | 6.1–12.6 | 9.4 ± 3.3 | 42–55 | 48.5 ± 6.5 |

**Table 1.** *Cont.*

| Species | Region | Stations | Numbers | Year | Month | Body Weight/g | | Mantle Length/mm | |
|---|---|---|---|---|---|---|---|---|---|
| | | | | | | Range | Mean | Range | Mean |
| *Uroteuthis duvaucelii* | ECS | NB | 307 | 2019 2020 | 11 1, 9–11 | 3.3–66.7 | 25.2 ± 13.7 | 37–118 | 79.2 ± 18.5 |
| | | ND | 293 | 2019 2020 | 11, 12 8, 9, 11 | 2.1–51.5 | 20.7 ± 11.1 | 28–124 | 77.7 ± 25.3 |
| | NSCS | ZZ | 764 | 2019 2020 | 8–12 1, 4–8 | 1.5–57.5 | 15.5 ± 10.3 | 28–132 | 69.0 ± 19.3 |
| | | ST | 840 | 2019 2020 | 8, 10–12 1, 4–11 | 2.4–63.1 | 12.4 ± 7.3 | 34–121 | 61.6 ± 11.6 |
| | BBG | YJ | 411 | 2020 | 5–11 | 12.9–82.0 | 33.4 ± 8.8 | 65–138 | 99.1 ± 10.7 |
| | | BH | 778 | 2019 2020 | 7, 9, 12 1, 4–11 | 3.6–138.0 | 16.3 ± 9.38 | 40–170 | 69.1 ± 14.3 |
| | | HK | 108 | 2019 | 7, 8, 10 | 16.4–92.9 | 38.9 ± 16.9 | 68–145 | 96.2 ± 17.2 |
| | | DZ | 100 | 2020 | 6 | 18.6–160.9 | 80.0 ± 38.0 | 60–215 | 143.5 ± 41.9 |
| | | LG | 50 | 2020 | 5 | 27.6–125.7 | 67.7 ± 20.3 | 99–171 | 120.5 ± 13.2 |

Note: NYS: North Yellow Sea; CYS: Central Yellow Sea; SYS: Southern Yellow Sea; ECS: East China Sea; NSCS: Northern South China Sea; BBG: Beibu Gulf.

To ensure normal distribution, the mantle length and body weight data were log-transformed, and SST data were mean-centered [8]. Mixed effects models were employed to analyze the monthly body weight variation patterns of the two species. A generalized linear model (GLM), a simple linear model suitable for individuals of different months and regions, and four linear mixed effect models (LMEM) that consider mouth as random factors to explain the relationship between *BW* and *ML* were fitted (Table 2) [42]. The optimal weight–growth model was determined based on Akaike's information criterion for small sample size (AICc), which was corrected for the small sample size [43]. The variance in weight growth explained by the model was evaluated using the condition $R^2$ metric [44]. Finally, external influences such as SST were considered as fixed effects to analyze their impact on body weight. All analyses were conducted using the RStudio 3.5 program, with the 'lme4' package employed for building mixed models, the 'MuMIn' package used to calculate conditional $R^2$-values, and the 'AICcmodavg' package employed to calculate AICc values.

**Table 2.** Models for length–weight relationships of *Loligo beka* and *Uroteuthis duvaucelii*.

| Model Factors | *Loligo beka* | | | *Uroteuthis duvaucelii* | | |
|---|---|---|---|---|---|---|
| | *df* | *AICc* | $R^2$ | *df* | *AICc* | $R^2$ |
| $\ln(BW) = \ln(a) + b \times \ln(ML)$ | 3 | −290.08 | 0.84 | 3 | −1657 | 0.92 |
| $\ln(BW) = \ln(a) + b \times \ln(ML) +$ Month | 4 | −403.49 | 0.85 | 4 | −1732 | 0.93 |
| $\ln(BW) = \ln(a) + b \times \ln(ML) +$ Sex + Month | 5 | −532.64 | 0.86 | 5 | −1997 | 0.93 |
| $\ln(BW) = \ln(a) + b \times \ln(ML) +$ Sex + lnML × Sex + Month | 6 | −528.19 | 0.86 | 6 | −1999 | 0.93 |
| $\ln(BW) = \ln(a) + b \times \ln(ML) +$ Sex + lnML × Sex + Region + Month | 7 | −692.25 | 0.87 | 7 | −2068 | 0.94 |

## 3. Result

### 3.1. Mantle Length and Body–Weight Distribution

The frequency distribution of mantle length (ML) and body weight (BW) of *Loligo beka* and *Uroteuthis duvaucelii* vary latitudinally (Figures 2 and 3). The ML distribution of *L. beka* in NYS, CYS, and ECS regions is unimodal with dominant ML groups of 40–45, 50–55, and 45–50 mm, respectively. In *U. duvaucelii*, the ML distribution is unimodal in the ECS and BBG regions, with dominant ML groups of 70–80 mm, and bimodal in the NSCS region, with peaks of 60–70 mm and 90–100 mm. The BW distribution of *L. beka* is concentrated between 5 and 10 g in all 3 regions, with the ECS region having the largest proportion

(54.44%). In contrast, the BW distribution of *U. duvaucelii* is concentrated between 10 and 20 g in all 3 regions, with the BBG region having the largest proportion (45.81%). The maximum range and mean values of ML and BW for *L. beka* and *U. duvaucelii* are located in CYS and BBG regions, respectively.

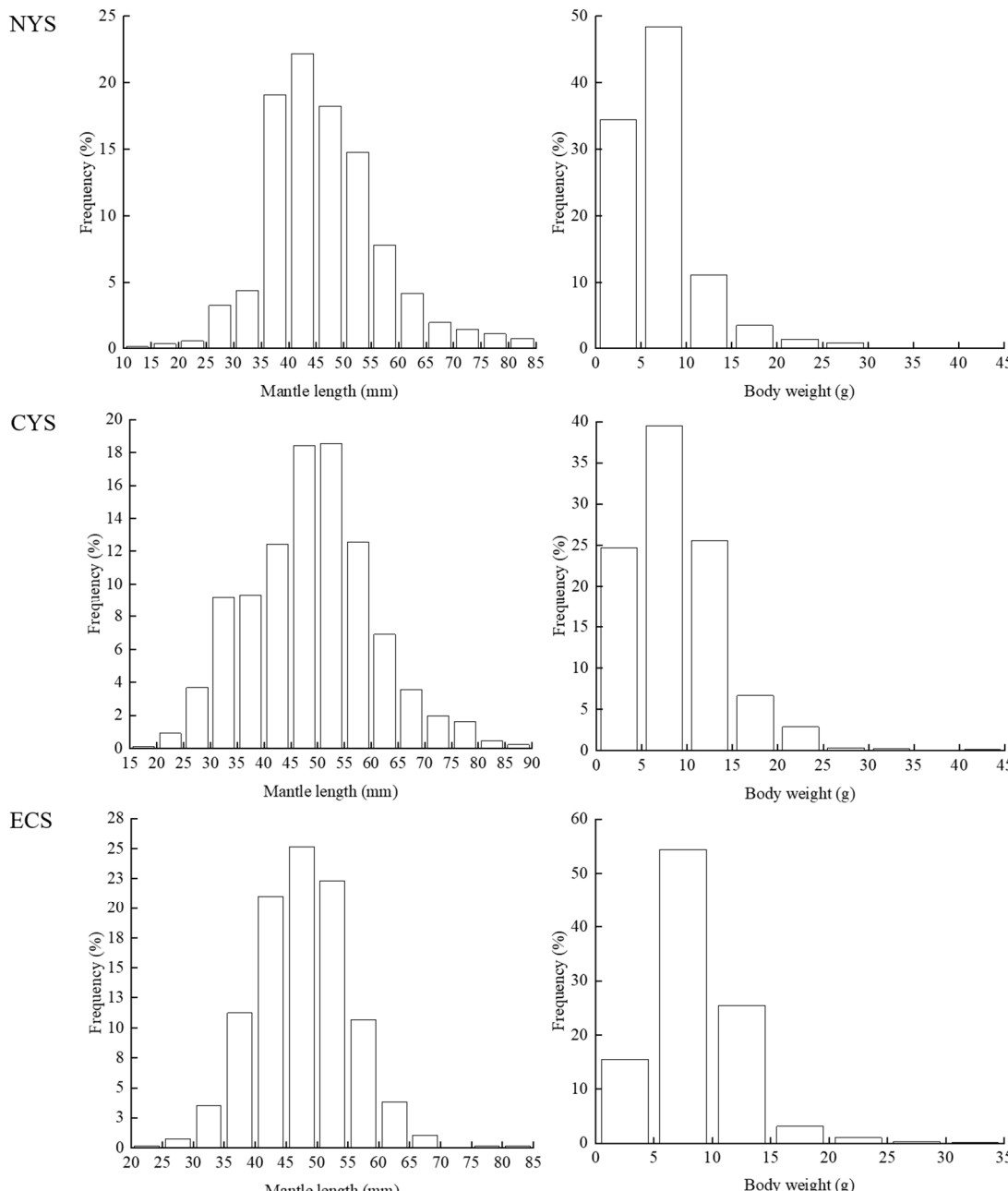

**Figure 2.** Frequency distribution of mantle length and body weight of *Loligo beka* (NYS: Northern Yellow Sea; CYS: Central Yellow Sea; ECS: East China Sea).

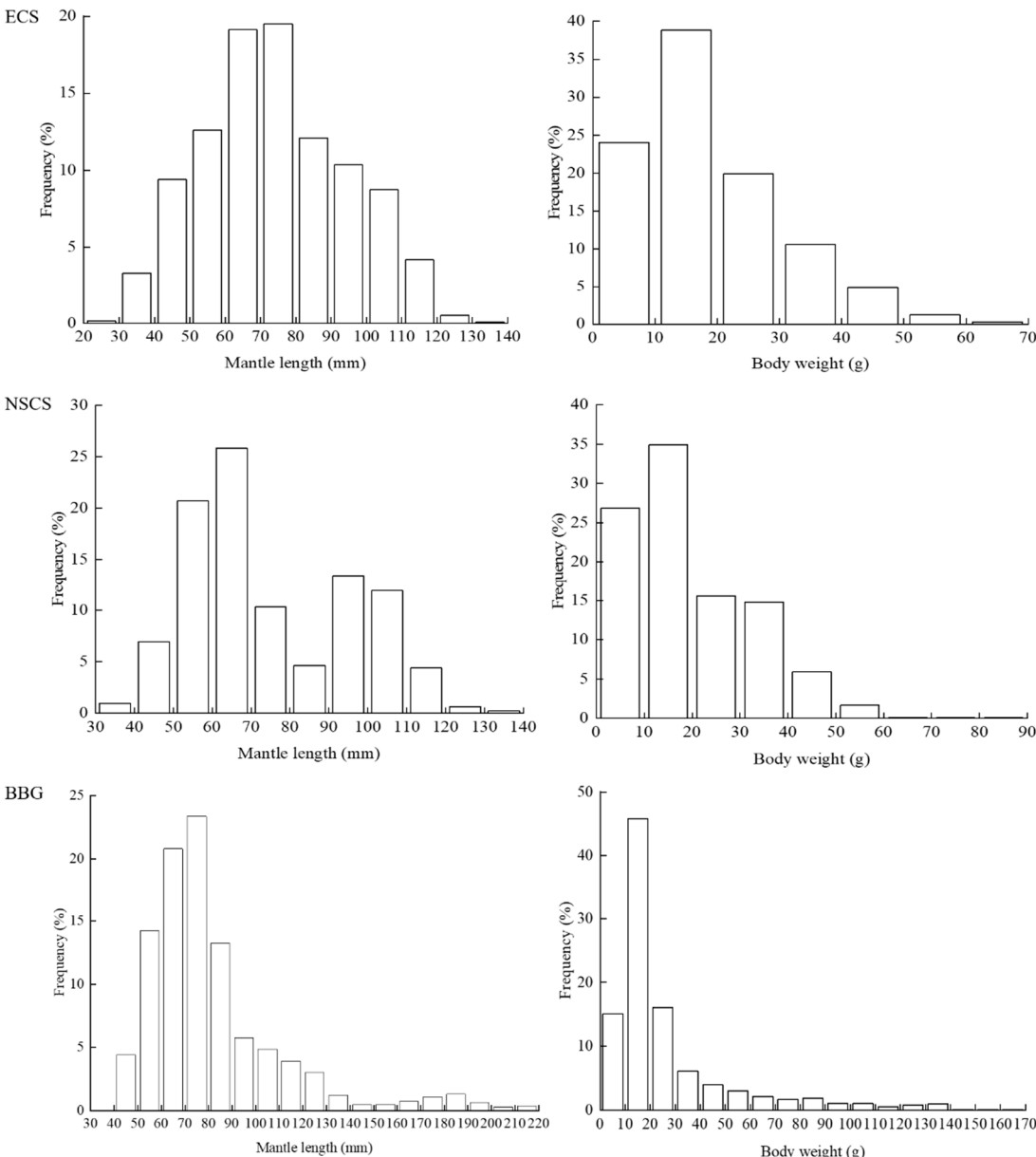

**Figure 3.** Frequency distribution of mantle length and body weight of *Uroteuthis duvaucelii* (ECS: East China Sea; NSCS: Northern South China Sea; BBG: Beibu gulf).

*3.2. Length–Weight Relationship*

The regional variation in the length–weight relationship (LWR) of *Loligo beka* and *Uroteuthis duvaucelii* is shown in Figure 4. The *b* values of these species in each area are less than 3, indicating allometric growth throughout the year. Tables 3 and 4 show that significant differences exist in growth patterns between males and females and between different latitudes. The LWR of individuals in each region of the 2 species exhibited significant monthly changes (as shown in Figures 5 and 6), and the values of parameter *b* varied between regions (as detailed in Tables 3 and 4). Differences in the LWR of these species between and within regions, including seasonal and latitudinal differences, are primarily influenced by the environmental conditions of the respective areas. This reflects the two Loliginidae squid species' strong environmental adaptability and the formation of different geographical groups.

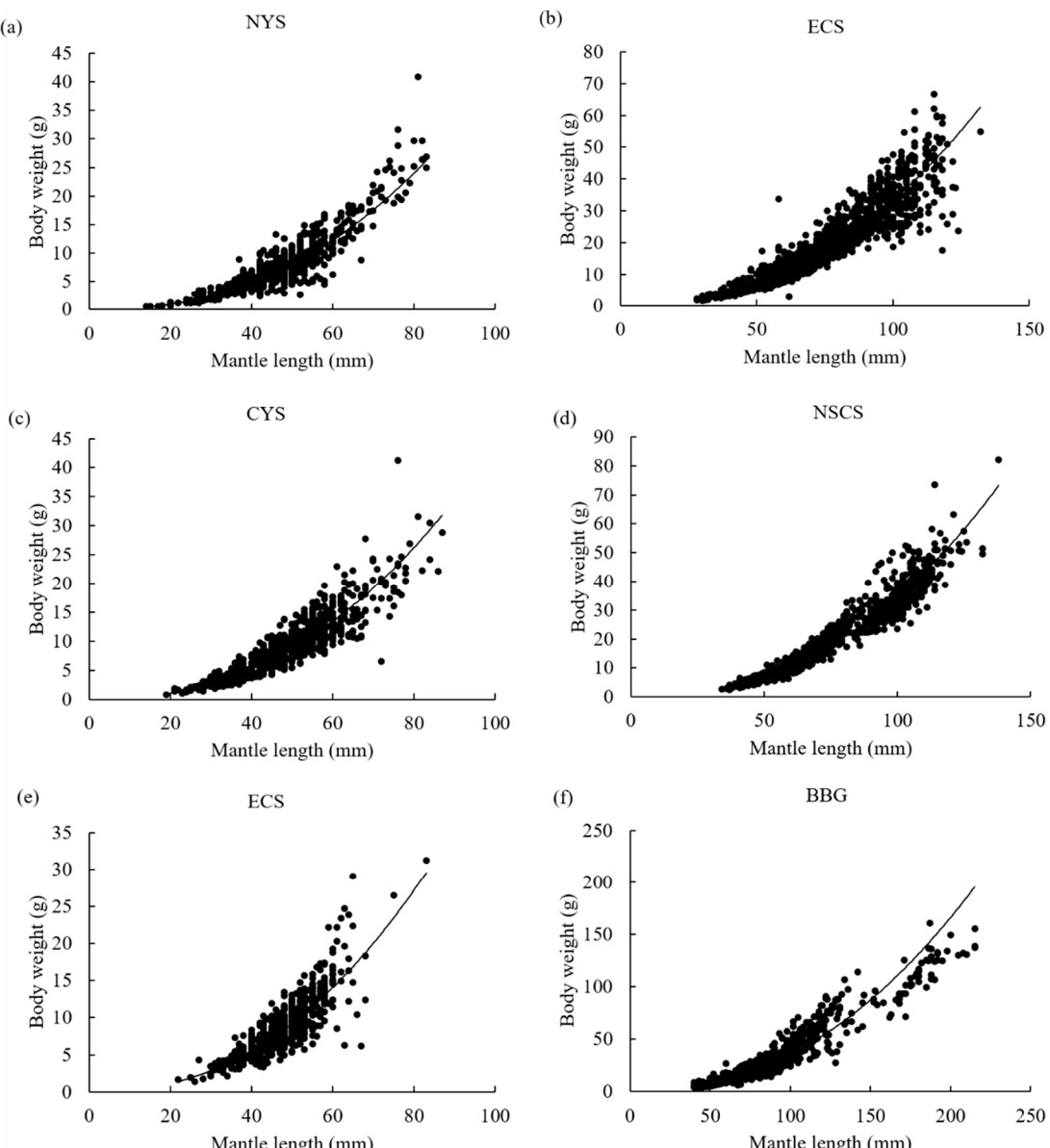

**Figure 4.** Regional differences in the relationship between mantle length and body weight for *Loligo beka* (**a**,**c**,**e**) and *Uroteuthis duvaucelii* (**b**,**d**,**f**) (NYS: Northern Yellow Sea; CYS: Central Yellow Sea; ECS: East China Sea; NSCS: Northern South China Sea; BBG: Beibu Gulf).

### 3.3. Variation in Weight Growth across Latitudes for Loligo beka and Uroteuthis duvaucelii

The present study examined the variation in body weight across latitudes for *Loligo beka* and *Uroteuthis duvaucelii*. Linear mixed effect models with month and region as random effects on the intercept were found to be the best model, as determined by Akaike's information criterion for small sample sizes (Table 2). The models explained 87% and 94% of variance for *L. beka* and *U. duvaucelii*, respectively. The results revealed that body weight varied greatly among months, and intra-specific weight patterns differed among regions, with two main peaks observed in different regions. This suggests that the condition factor of these species exhibits noticeable differences across latitudes (Figure 7). Furthermore, the body weight of the two species in the ECS were similar from January to July, but differed significantly from August to December, indicating the growth differences among species and the life cycle characteristics of seasonal habitats in the regional environment synchronization. The effect of the "region" factor was found to play a crucial role In

understanding the condition factor changes across latitudes of the two species, while the "month" factor had a local effect within the current year.

**Table 3.** Power function equation parameters of the fitting relationship between mantle length and body weight of *Loligo beka*.

| Areas | Sex | Month | N | $a$ | $b$ | $R^2$ |
|---|---|---|---|---|---|---|
| Northern Yellow Sea | All | All | 933 | 0.0009 | 2.3363 | 0.8537 |
| | Male | All | 682 | 0.0014 | 2.2022 | 0.7590 |
| | Female | All | 251 | 0.0010 | 2.3179 | 0.8658 |
| | All | 1 | 60 | 0.0066 | 1.7739 | 0.7634 |
| | All | 4 | 60 | 0.0035 | 1.9839 | 0.8581 |
| | All | 5 | 60 | 0.0147 | 1.5506 | 0.5202 |
| | All | 6 | 92 | 0.0013 | 2.1542 | 0.4696 |
| | All | 7 | 131 | 0.0004 | 2.5229 | 0.8908 |
| | All | 8 | 60 | 0.0032 | 2.0188 | 0.7813 |
| | All | 9 | 190 | 0.0015 | 2.2192 | 0.7165 |
| | All | 10 | 100 | 0.0002 | 2.8383 | 0.8708 |
| | All | 11 | 60 | 0.0177 | 1.4937 | 0.4864 |
| | All | 12 | 120 | 0.0020 | 2.1009 | 0.6901 |
| Central Yellow Sea | All | All | 868 | 0.0010 | 2.3269 | 0.8166 |
| | Male | All | 608 | 0.0011 | 2.2920 | 0.7748 |
| | Female | All | 260 | 0.0014 | 2.2571 | 0.7846 |
| | All | 1 | 60 | 0.0063 | 1.8498 | 0.5441 |
| | All | 4 | 60 | 0.0006 | 2.4257 | 0.6965 |
| | All | 6 | 114 | 0.0013 | 2.2362 | 0.7311 |
| | All | 7 | 204 | 0.0009 | 2.3381 | 0.8297 |
| | All | 8 | 130 | 0.0002 | 2.7131 | 0.7609 |
| | All | 9 | 130 | 0.0004 | 2.6210 | 0.8910 |
| | All | 10 | 50 | 0.0007 | 2.3745 | 0.7791 |
| | All | 11 | 60 | 0.0003 | 2.6568 | 0.7957 |
| | All | 12 | 60 | 0.0031 | 2.0699 | 0.8244 |
| East China Sea | All | All | 676 | 0.0012 | 2.2957 | 0.7306 |
| | Male | All | 439 | 0.0025 | 2.0792 | 0.6619 |
| | Female | All | 237 | 0.0018 | 2.2003 | 0.6786 |
| | All | 1 | 60 | 0.0013 | 2.2874 | 0.9103 |
| | All | 4 | 60 | 0.0138 | 1.6969 | 0.5453 |
| | All | 5 | 60 | 0.0012 | 2.2890 | 0.6969 |
| | All | 6 | 60 | 0.0105 | 1.7204 | 0.6111 |
| | All | 7 | 60 | 0.0042 | 1.9954 | 0.7792 |
| | All | 8 | 146 | 0.0038 | 1.9612 | 0.5448 |
| | All | 9 | 120 | 0.0008 | 2.4053 | 0.7911 |
| | All | 10 | 50 | 0.0011 | 2.2671 | 0.7593 |
| | All | 12 | 60 | 0.0007 | 2.4402 | 0.8686 |

*3.4. Temperature Effect on Weight Growth*

The impact of sea surface temperature (SST) on body weight of *Loligo beka* and *Uroteuthis duvaucelii* was investigated, and a significant correlation was observed for both species. Randomization tests indicated a $P_{\Delta AICc}$ of less than 0.001, signifying that the response of body weight to SST is statistically significant. The findings suggest that as SST increases, body weight also increases, as demonstrated by the relationship between SST and body weight illustrated in Figure 8.

**Table 4.** Power function equation parameters of the fitting relationship between mantle length and body weight of *Uroteuthis duvaucelii*.

| Areas | Sex | Month | N | *a* | *b* | $R^2$ |
|---|---|---|---|---|---|---|
| East China Sea | All | All | 1364 | 0.0010 | 2.2624 | 0.9000 |
| | Male | All | 1190 | 0.0009 | 2.2868 | 0.8552 |
| | Female | All | 174 | 0.0165 | 1.6519 | 0.7499 |
| | All | 1 | 180 | 0.0018 | 2.1305 | 0.7200 |
| | All | 4 | 60 | 0.0005 | 2.4799 | 0.8659 |
| | All | 5 | 60 | 0.0008 | 2.2859 | 0.9149 |
| | All | 6 | 60 | 0.0012 | 2.2054 | 0.8609 |
| | All | 7 | 84 | 0.0031 | 2.0187 | 0.8883 |
| | All | 8 | 150 | 0.0002 | 2.6204 | 0.7504 |
| | All | 9 | 153 | 0.0001 | 2.7474 | 0.9401 |
| | All | 10 | 110 | 0.0001 | 2.9671 | 0.8073 |
| | All | 11 | 367 | 0.0025 | 2.0539 | 0.7686 |
| | All | 12 | 140 | 0.0005 | 2.4164 | 0.9259 |
| Northern South China Sea | All | All | 1251 | 0.0008 | 2.3301 | 0.9556 |
| | Male | All | 1149 | 0.0008 | 2.3167 | 0.9359 |
| | Female | All | 102 | 0.0007 | 2.3547 | 0.9403 |
| | All | 1 | 60 | 0.0002 | 2.6819 | 0.9208 |
| | All | 4 | 60 | 0.0019 | 2.1194 | 0.9120 |
| | All | 5 | 120 | 0.0004 | 2.4956 | 0.9322 |
| | All | 6 | 103 | 0.0004 | 2.4508 | 0.9786 |
| | All | 7 | 120 | 0.0018 | 2.1289 | 0.9553 |
| | All | 8 | 128 | 0.0003 | 2.5403 | 0.9599 |
| | All | 9 | 120 | 0.0032 | 1.9930 | 0.9522 |
| | All | 10 | 170 | 0.0013 | 2.2080 | 0.9686 |
| | All | 11 | 310 | 0.0003 | 2.5018 | 0.8800 |
| | All | 12 | 60 | 0.0011 | 2.2538 | 0.8399 |
| Beibu Gulf | All | All | 1036 | 0.0012 | 2.2328 | 0.9224 |
| | Male | All | 836 | 0.0018 | 2.1354 | 0.9594 |
| | Female | All | 200 | 0.0011 | 2.2969 | 0.8065 |
| | All | 1 | 60 | 0.0275 | 1.4695 | 0.7084 |
| | All | 4 | 60 | 0.0004 | 2.4517 | 0.9161 |
| | All | 5 | 110 | 0.0001 | 2.7653 | 0.8691 |
| | All | 6 | 137 | 0.0037 | 1.9915 | 0.9435 |
| | All | 7 | 191 | 0.0014 | 2.1817 | 0.8318 |
| | All | 8 | 68 | 0.0015 | 2.1724 | 0.9031 |
| | All | 9 | 120 | 0.0019 | 2.1568 | 0.6975 |
| | All | 10 | 110 | 0.0005 | 2.4722 | 0.8936 |
| | All | 11 | 120 | 0.0011 | 2.2393 | 0.7669 |
| | All | 12 | 60 | 0.0023 | 2.0667 | 0.7075 |

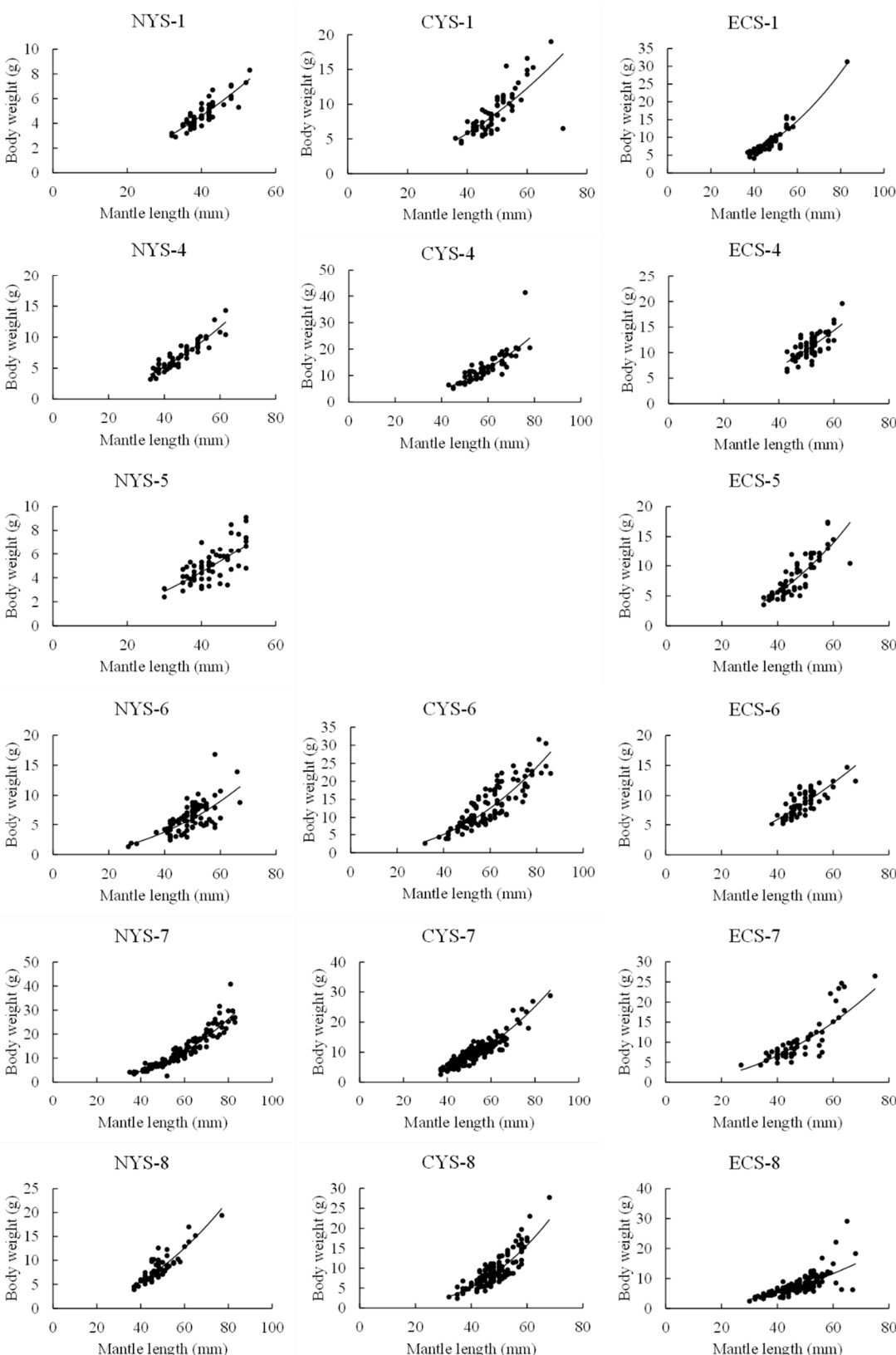

**Figure 5.** *Cont.*

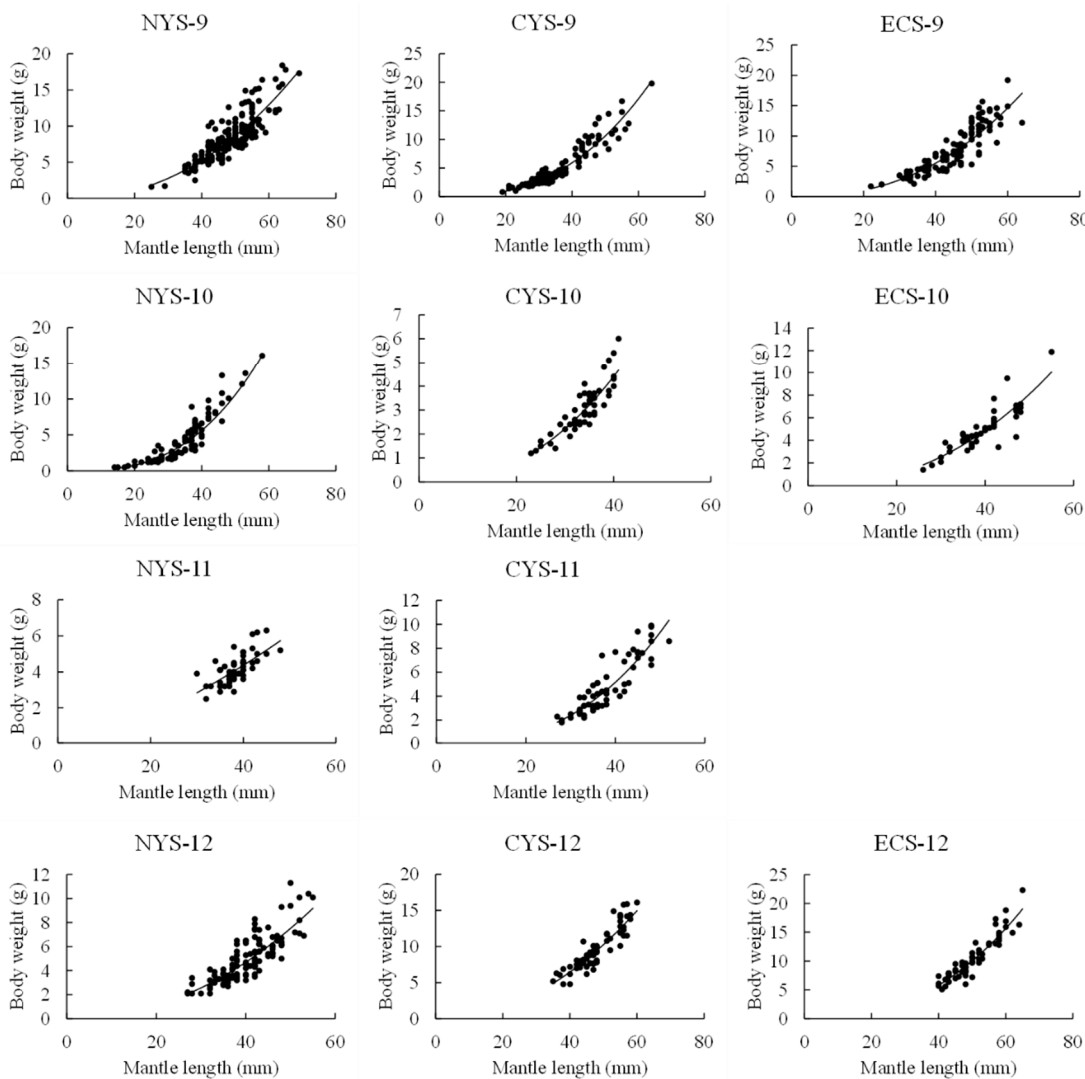

**Figure 5.** Monthly variation of fitting relationship between mantle length and body weight in different areas for *Loligo beka* (NYS: Northern Yellow Sea; CYS: Central Yellow Sea; ECS: East China Sea).

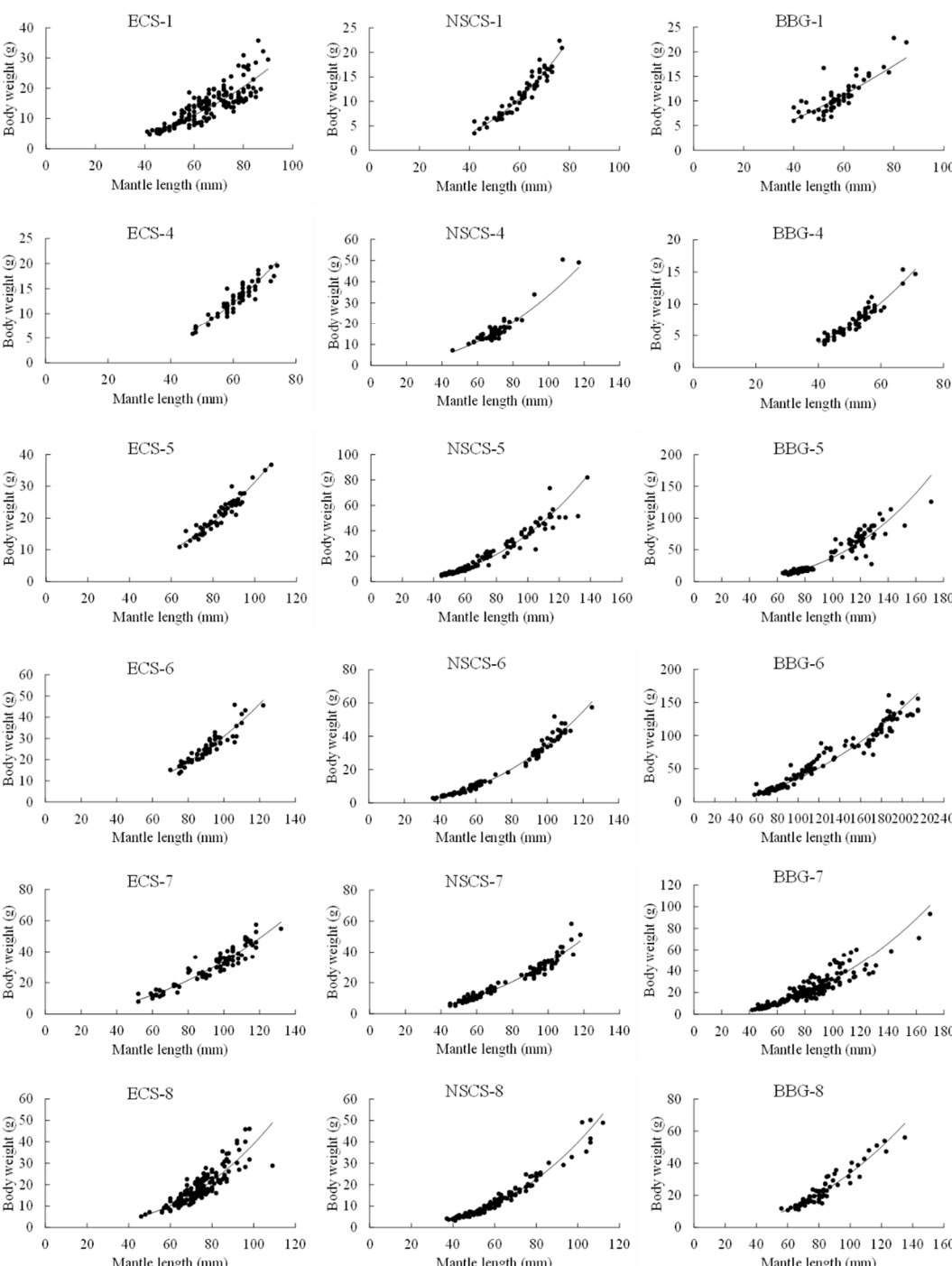

**Figure 6.** *Cont.*

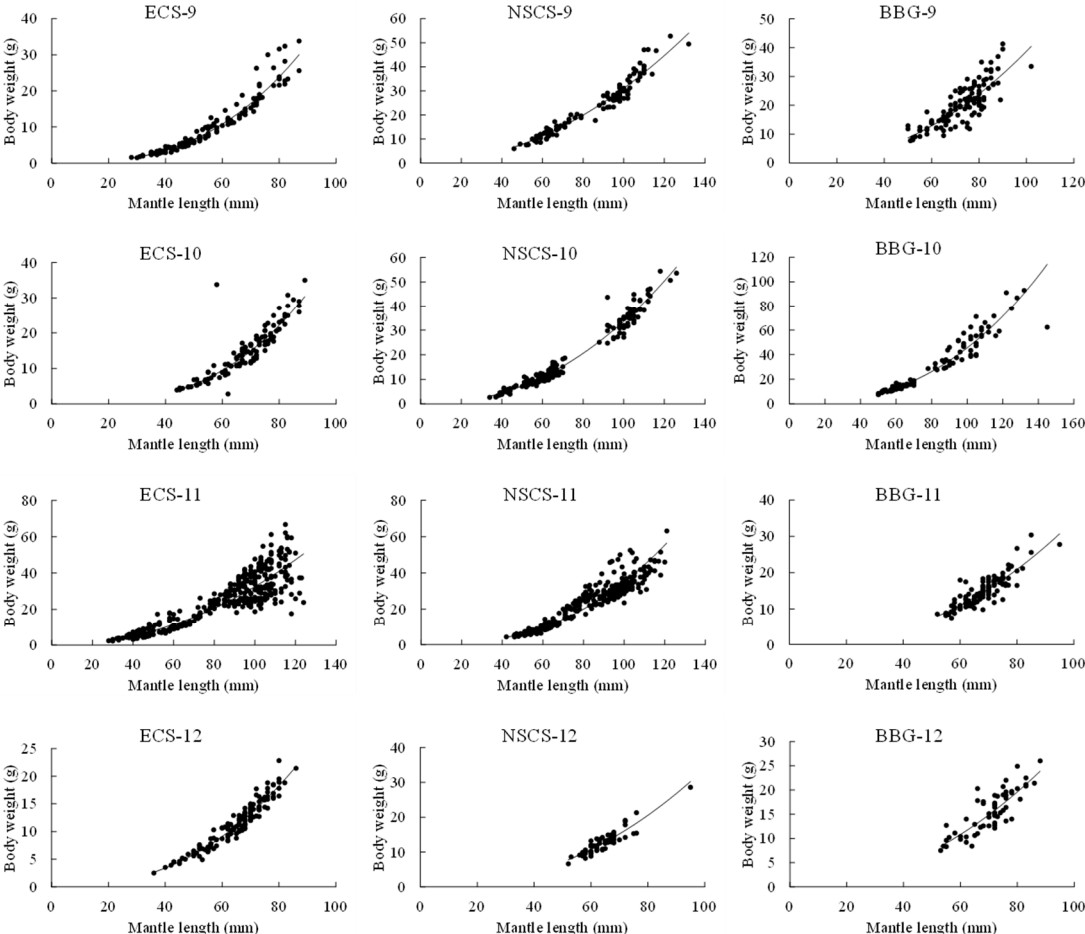

**Figure 6.** Monthly variation of fitting relationship between mantle length and body weight in different areas for *Uroteuthis duvaucelii* (ECS: East China Sea; NSCS: Northern South China Sea; BBG: Beibu Gulf).

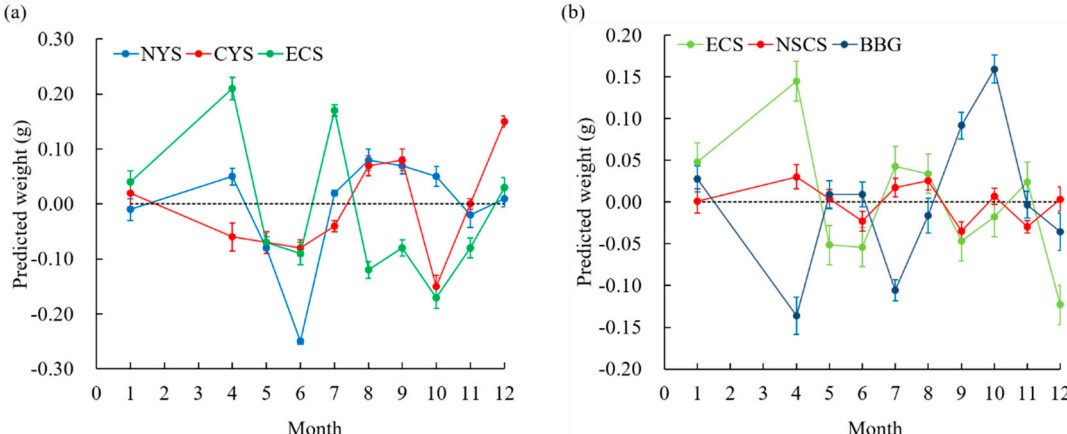

**Figure 7.** Monthly variation of condition factor across latitudes based on month random effect estimates (±SE) for *Loligo beka* (**a**) and *Uroteuthis duvaucelii* (**b**) (NYS: Northern Yellow Sea; CYS: Central Yellow Sea; ECS: East China Sea; NSCS: Northern South China Sea; BBG: Beibu Gulf).

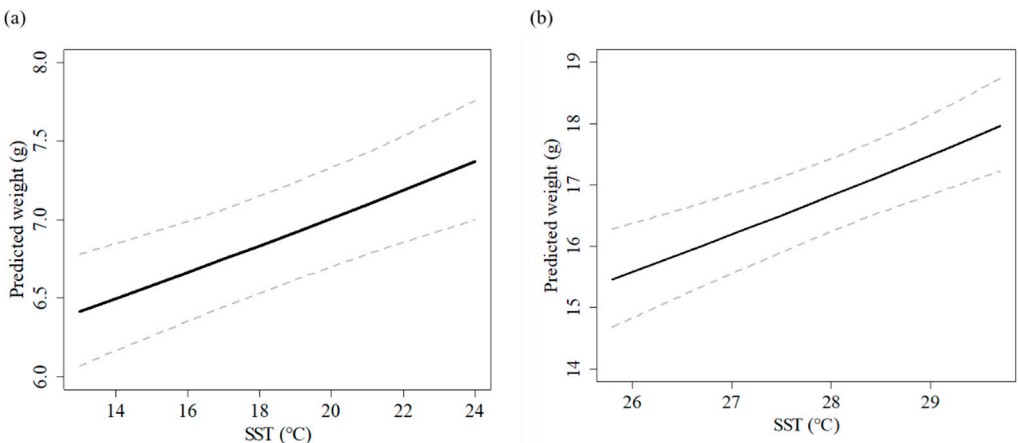

**Figure 8.** Predicted effects of SST on weight for *Loligo beka* (**a**) and *Uroteuthis duvaucelii* (**b**). The gray dotted line represents the 95% confidence interval, and the black solid line represents the fitting relationship between SST and weight.

## 4. Discussion

The growth of cephalopods is affected by various factors, including life stages, gonad development, food supply, and habitat environments [13,45]. Assessing growth is an ideal parameter to determine the individual response to environmental changes and fishing [46], and differences in population phenotypic parameters can identify separate management populations to optimize yield [47,48]. In this study, samples were collected monthly from the main distribution areas of *L. beka* and *U. duvaucelii* in the China Seas to understand their growth patterns and population dynamics in response to environmental warming.

The results showed significant growth differences between sexes, indicating sexual dimorphism within the species, and the same phenomenon has also been observed in *Uroteuthis edulis*, *Uroteuthis chinensis*, and *Loligo duvauceli* [49,50]. Both species demonstrated significant inter-month growth changes with two spawning peaks (Figure 7). *L. beka* colonizes for reproductive migration in spring and lays more eggs in the inner bay [20]; the breeding pattern of *L. beka* is similar to that of *L. japonica*, and spawning peak is from April to June [51,52]; its breeding populations are more obvious in spring and autumn, and the clusters are dense and large in number from September to October [18]. *U. duvaucelii*, on the other hand, breeds almost year-round, with peaks occurring mainly in spring and autumn [21]. The reproductive patterns observed in *L. beka* and *U. duvaucelii* are consistent with other Loliginidae squids in the China Seas, such as *U. edulis* and *U. chinensis* [19,27,53,54]. These findings provide valuable information for understanding the growth patterns and population dynamics of cephalopods and optimizing yield through separate management populations.

The present study examined the condition factor of two species of Loliginidae squids, *L. beka* and *U. duvaucelii*, with respect to their growth patterns, environmental adaptations, and geographical distribution. The condition factor of these species showed significant latitudinal differences, which were attributed to local environmental adaptability, particularly temperature, since lifespan changes both within and between species are related to ambient temperature [2]. In a wide geographical range, squid growth has great plasticity with changes in temperature [15]. The study found that temperature had a positive linear growth relationship with these species, which was consistent with previous studies on other Loliginid squid species, such as *L. forbesi*, *L. pealeii*, and *L. japonica* [11,55,56]. Furthermore, both species showed similar body weight patterns in the East China Sea (ECS) from January to July (Figure 7), possibly because both inhabit shallow waters in this region [57]. The study also found that the dominant population group (5–10 g) of *L. beka* in the ECS accounted for the largest proportion (Figures 2 and 3), whereas that for *U. duvaucelii* (10–20 g) in the BBG accounted for the largest proportion. The study suggests that the higher proportion of small-sized individuals in lower latitudes is due to the warm waters

that support a higher metabolic rate and higher food intake, resulting in rapid biological growth, and the community composition is dominated by small-sized individuals [45], which provides a biological explanation for Loliginidae squid growth with the increase in temperature, and better fatness means higher survival rate and fecundity.

The study further examined the geographical distribution of the two species and found that *Uroteuthis duvaucelii* was mainly distributed in the East China Sea and Northern South China Sea (Table 1), indicating that its dominant position has not changed [17]. In contrast, the main capture area for *L. beka* was in the Yellow Sea (YS) and ECS—and its population structure—may have changed due to the fishing nets and strategies employed by local fishermen. *L. beka* is mainly caught in set and trawl nets as by-catch [19,24], whereas *U. duvaucelii*, *U. edulis*, and *U. chinensis* are the dominant species in the NSCS and have a high economic impact, especially *U. duvaucelii* and *U. chinensis*, making them a major target for local fishermen [20,23]. The study suggests that the fishing nets dominated by these major economic species are detrimental to the capture of small-sized *L. beka*. Previous studies have shown that *L. beka* is one of the important commercial economic cephalopod species in China, mainly caught in set and trawl nets as by-catch [19,24]. For example, *L. beka* was captured in four seasonal bottom-trawl fisheries surveys in offshore of NSCS from 2014 to 2015, but it is not the dominant species in this area [23]. There were obvious latitude differences in the growth of *L. beka* and *U. duvaucelii*, which provided important scientific support for the management and development of Loliginid squid resources in the future.

The management of cephalopod populations is a complex issue due to their unique life history characteristics, including short lifespans, rapid growth, high natural mortality, and complex population structures [2]. Environmental conditions have a significant impact on population structure [58], leading to spatial–temporal variability, particularly for exploited squid species [59,60]. Population research necessitates the integration of multiple methods to draw reliable conclusions, especially for short-lived cephalopods. Age identification and growth characteristics research methods are of great significance in identifying, characterizing, and estimating population distribution, resource dynamics, life history, and fishery sustainability for cephalopod species [35,61]. This study examines the latitudinal difference in condition factor between *L. beka* and *U. duvaucelii* and the effect of temperature on their body weight growth in the China Seas. The results indicate significant latitude differences in weight growth of the two species and their consistent response to temperature, which confirms their environmental adaptability and supports the use of Loliginidae squids as indicator species for studying environmental change in the China Seas. These findings provide valuable insight into the growth pattern and response to environmental changes of *L. beka* and *U. duvaucelii* in the China Seas and contribute to the understanding and management of cephalopod resources in the region. Future studies will focus on age identification of the two species using the statolith to gain an in-depth understanding of their growth differences.

**Author Contributions:** Conceptualization, C.Z. and Y.T.; Data Curation, C.Z. and Y.T.; Methodology, C.Z.; Funding Acquisition, Y.T.; Software, J.G.; Formal Analysis, J.G.; Investigation, J.G., Z.L. and D.L.; Writing—Original Draft Preparation, J.G.; Writing—Review & Editing, C.Z. and Y.T.; Visualization, J.G., C.Z. and Y.T. All authors have read and agreed to the published version of the manuscript.

**Funding:** This research was supported by the National Natural Science Foundation of China (No. 41930534).

**Institutional Review Board Statement:** Not applicable.

**Acknowledgments:** The authors thank Haozhan Wang for his help during the sample collection. We also thank the editors and anonymous reviewers for helping to improve the manuscript.

**Conflicts of Interest:** The authors declare no conflict of interest.

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
