# Peer review of "Latitudinal Difference in the Condition Factor of Two Loliginidae Squid (Beka Squid and Indian Squid) in China Seas"

_diversity, doi:10.3390/d15070812_

Round 1

Reviewer 1 Report

Main comments:

I think this work is very interesting and should be published, especially in light of the great importance of the ecological and economic impact of the two species.

I suggest only a few reviews to the Authors to improve the exposure of the text.

Suggestions

Materials and Methods.

Table 1 (Sample data): The column for "sample time" is unclear.

I recommend to the Authors insert two separate columns: "year" and "month" after the column "stations". This way makes the information easier to interpret.

 Discussion :

lines 360-373: I suggest reviewing the discussions:

Particularly about “multiple methods” mentioned : what about “Age determination”?

It could be very interesting to estimate the age of these species and to compare the data obtained with this work. We know very well that hard structures such as statoliths, are good tools for dating cephalopods, especially in Loliginidae squids, and that they can "record" environmental changes, such as changes in temperature, Ph, etc." I suggest the authors increase the text of the conclusions by inserting some age considerations, and I also suggest mentioning the recent paper on the age investigation of two Loliginidae species Loligo vulgaris and Loligo forbesii through the study of the lens (“Age determination of Loligo vulgaris and Loligo forbesii using eye lens analysis DOI:10.1007/s00435-017-0381-8) as a possible future investigation for the two species L. beka and U. duvaucelii.

Author Response

Dear Reviewers,

Thanks for your comments concerning our manuscript. Those comments are all valuable and very helpful for revising and improving our paper, as well as the important guiding significance to our researches. We have studies comments carefully and have made the corrections. We hope it meets with approval. The main corrections in the paper and the responds to your comments are as follows.

Kind regards

Jianzhong Guo

Author Responses:

Point 1: Table 1 (Sample data): The column for "sample time" is unclear.

I recommend to the Authors insert two separate columns: "year" and "month" after the column "stations". This way makes the information easier to interpret.

Response 1: Thank for this valuable suggestion. We accepted it and added as suggested. (Please see Page 5, Table 1).

Point 2:  Discussion : lines 360-373: I suggest reviewing the discussions:

Particularly about “multiple methods” mentioned : what about “Age determination”?

It could be very interesting to estimate the age of these species and to compare the data obtained with this work. We know very well that hard structures such as statoliths, are good tools for dating cephalopods, especially in Loliginidae squids, and that they can "record" environmental changes, such as changes in temperature, Ph, etc." I suggest the authors increase the text of the conclusions by inserting some age considerations, and I also suggest mentioning the recent paper on the age investigation of two Loliginidae species Loligo vulgaris and Loligo forbesii through the study of the lens (“Age determination of Loligo vulgaris and Loligo forbesii using eye lens analysis DOI:10.1007/s00435-017-0381-8) as a possible future investigation for the two species L. beka and U. duvaucelii.

Response 2: Thank for this valuable suggestion. We accepted it and added as suggested. (Please see Page 18,lines364-366; 375-376).

References:

  1. Agus, B., Mereu, M., Cannas, R., Cau, A., Coluccia, E.,Follesa, M. C. and Cuccu, D. Age determination of Loligo vulgaris and Loligo forbesii using eye lens analysis. Zoomorphology2018,137(1), 63-70. 
  2. Goicochea-Vigo, C., Morales-Bojórquez, E., Zepeda-Benitez, V. Y., and Suclupe, D. A. Age and growth estimates of the jumbo flying squid (Dosidicus gigas) off Peru. Aquatic Living Resources, 2019, 32,

Reviewer 2 Report

This is timely and important manuscript in the field. My main concern is the level of English, and thus the Authors are strongly encouraged to have their manuscript proof-read by a native speaker. There are also some minor inconsistencies and suggested corrections:

Line 24: how can species be allometric? Growth is allometric, not species;

Line 29: ‘habitat environment’ is not a good English expression and needs to be rephrased;

Line 40: the Authors are advised to update the references to more recent ones;

Line 48: ‘they’ fits better here;

Line 72 and throughout the manuscript: it’s better not to start a sentence from abbreviation;

Line 90: growth status of the squids, not of the sea itself as written here;

Line 185, Table 2: the Authors are advised to add p values to the table;

Line 359: reference cited out of style for the manuscript.

See above: the Authors are strongly encouraged to have their manuscript proof-read by a native speaker.

Author Response

Dear Reviewers,

Thanks for your comments concerning our manuscript. Those comments are all valuable and very helpful for revising and improving our paper, as well as the important guiding significance to our researches. We have studies comments carefully and have made the corrections. We hope it meets with approval. The main corrections in the paper and the responds to your comments are as follows.

Kind regards

Jianzhong Guo

Author Responses:

Point 1: Line 24: how can species be allometric? Growth is allometric, not species; 

Response 1: Thanks the reviewer for the careful examination. We agreed it and added the word "growth" as suggested. The detailed amendment has been presented in the revised manuscript (Please see Page 1, Line 23).

Point 2: Line 29: ‘habitat environment’ is not a good English expression and needs to be rephrased;

Response 2: Thanks the reviewer for the careful examination. We accepted and replaced “habitat environment” with “natural habitats” as suggested. The detailed amendment has been presented in the revised manuscript (Please see Page 1, Line 28).

Point 3: Line 40: the Authors are advised to update the references to more recent ones;

Response 3:Thank for this valuable suggestion. We accepted it and added as suggested. (Please see Page 19,lines393-397).

References:

  1. Luna, A., Sánchez, , Chicote, C., and Gazo, M. Cephalopods in the diet of Risso's dolphin (Grampus griseus) from the Mediterranean Sea: A review. Marine Mammal Science, 2022, 38, 725-741.
  2. Arkhipkin, A. I., Hendrickson, L. C., Ignaccio, , Pierce, G. J., Roa-Ureta, R. H., Jran-Paul, R. and Andreas, W. Stock assessment and management of cephalopods: advances and challenges for short-lived fishery resources. ICES Journal of Marine Science. 2021, 78(2), 714-730.

Point 4: Line 48: ‘they’ fits better here;

Response 4:Thank for this valuable suggestion. We accepted and replaced “which” with “they” as suggested.  (Please see Page 1,line 44).

Point 5: Line 72 and throughout the manuscript: it’s better not to start a sentence from abbreviation;

Response 5:Thank for this valuable suggestion. We accepted and amended it as suggested.  The detailed amendment has been presented in the revised manuscript.

Point 6: Line 90: growth status of the squids, not of the sea itself as written here;

Response 6: Thanks the reviewer for the careful examination. We accepted and replaced “They” with “Growth status of the squids” as suggested. The detailed amendment has been presented in the revised manuscript (Please see Page 2, Lines 85-86).

Point 7: Line 185, Table 2: the Authors are advised to add p values to the table;

Response 7:Thank for this suggestion. The mixed effect model selected in this study adopts Akaike’s information criterion for small sample size (AICc) to evaluate the relative support of each candidate set of the model, corrected for the small sample size, and determined the optimal weight growth model of those two species. The condition R2 metric was used to evaluate the variance in weight growth explained by the model. AICc and R2 are used to select the optimal model, and p values are not used in this process. The significance of the effect of sea surface temperatures on body weight was analyzed based on the optimal model (p<0.001), as illustrated in Figure 8. (Please see Page 16, Line 290).

Point 8: Line 359: reference cited out of style for the manuscript.

Response 8: Thanks the reviewer for the careful examination. We accepted and amended it as suggested. The detailed amendment has been presented in the revised manuscript (Please see Pages 18, Line 361).

Point 9: Comments on the Quality of English Language

See above: the Authors are strongly encouraged to have their manuscript proof-read by a native speaker.

Response 9: Thank for this valuable suggestion. The language has been improved by a professional organization.

Reviewer 3 Report

Title: “Latitudinal difference in the growth…” the paper is not about the growth as squid age was not investigated. The correct title would be “Latitudinal differences in length-weight relations…” or in “condition factor” as on the line 159.

Line 52 – “and resulting in relatively complex populations” – unclear

Chapter 3.1 – such analysis makes no sense when maturity and sex are not taken into account, because of growth and sexual dimorphism. The authors should compare ML and BW of only mature squids, males and females separately.

Chapter 3.2 – the same for length-weight relation. Mature squids are perhaps 20-30% heavier at the same size than immature. Adult females are heavier than males of the same size. Authors should produce such graphs separately for immature, maturing and mature males and females.

Fig. 7 – again – it is not growth. Growth is an increase in length or weight of an individual or a group. The chapter 3.3 itself has little scientific value.

English language is sufficient, some minor improvements are desirable.

Author Response

Dear Reviewers,

Thanks for your comments concerning our manuscript. Those comments are all valuable and very helpful for revising and improving our paper, as well as the important guiding significance to our researches. We have studies comments carefully and have made the corrections. We hope it meets with approval. The main corrections in the paper and the responds to your comments are as follows.

Kind regards

Jianzhong Guo

Author Responses:

Point 1: Title: “Latitudinal difference in the growth…” the paper is not about the growth as squid age was not investigated. The correct title would be “Latitudinal differences in length-weight relations…” or in “condition factor” as on the line 159.

Response 1: Thank for this valuable suggestion. We accepted and replaced “growth” with “condition factor” as suggested. The detailed amendment has been presented in the revised manuscript (Please see Page 1, Line 2).

Point 2: Line 52 – “and resulting in relatively complex populations” – unclear

Response 2: Thanks the reviewer for the careful examination. We accepted and amended it as suggested. The detailed amendment has been presented in the revised manuscript (Please see Pages 2, Line 50).

Point 3: Chapter 3.1 – such analysis makes no sense when maturity and sex are not taken into account, because of growth and sexual dimorphism. The authors should compare ML and BW of only mature squids, males and females separately.

Response 3: Thank for this valuable suggestion. We have considered your suggestion before. However, due to the objective reasons of sample collection and the subjective reasons in the process of sample transportation and experimental analysis, as well as the fact that there are far more males than females and more mature individuals than immature individuals, the data could not reach statistical significance if male and female were analyzed separately in some months, this study finally studied the latitudinal difference in the condition factor of two Loliginidae squid from the whole region. Although there are certain deficiencies, large-scale sampling and data accumulation are basically carried out in the whole country. It is believed that this study will be of scientific value for understanding the biological characteristics and environmental adaptability of the main squid species in China Seas, and provide reference for the future management and sustainable development of Chinese cephalopod fishery resources.

Point 4: Chapter 3.2 – the same for length-weight relation. Mature squids are perhaps 20-30% heavier at the same size than immature. Adult females are heavier than males of the same size. Authors should produce such graphs separately for immature, maturing and mature males and females.

Response 4:Thank for this valuable suggestion. We have considered your suggestion before. However, due to the objective reasons of sample collection and the subjective reasons in the process of sample transportation and experimental analysis, as well as the fact that there are far more males than females and more mature individuals than immature individuals, the data could not reach statistical significance if male and female were analyzed separately in some months, this study finally studied the latitudinal difference in the condition factor of two Loliginidae squid from the whole region. Although there are certain deficiencies, large-scale sampling and data accumulation are basically carried out in the whole country. It is believed that this study will be of scientific value for understanding the biological characteristics and environmental adaptability of the main squid species in China Seas, and provide reference for the future management and sustainable development of Chinese cephalopod fishery resources.

Point 5: Fig. 7 – again – it is not growth. Growth is an increase in length or weight of an individual or a group. The chapter 3.3 itself has little scientific value.

Response 5: Thank for this valuable suggestion. We accepted and replaced “weight growth” with “condition factor”. (Please see Page 16,lines 274, 279, 283). 

However, we think the chapter 3.3 is very valuable. Condition factors are important for evaluating the health of aquatic species, populations, and communities (LeCren,1951; Ferreri, 2014). Without specific age data, condition factors have indicated biochemical, ecological, and physiological processes in studies on the phenotypic responses of  cephalopods to environmental change (Siddique et al., 2014) . In this study, weight growth was used as an indicator to evaluate the condition factor of those two species, the partial results reveal that condition factor of those two species present obvious latitude difference. The “region” factor effect plays a key role in understanding the condition factor changes across latitudes of the two species, while the “month” plays a local effect in the current year. These results help to understand the ecological characteristics of the two species in the absence of age data.

References:

LeCren, E. D. The length-weight relationship and seasonal cycle in gonad weight and condition in the Perch (Perca fluviatilis). Journal of Animal Ecology, 1951, 20(2), 201-219.

Ferreri, G. A. B. Length-weight relationships and condition factors of the Humboldt squid (Dosidicus gigas) from the Gulf of California and the Pacific Ocean. Journal of Shellfish Research, 2014, 33 (3), 769-780.

Siddique, M. A. M., Arshad, A., and Nurul Amin, S. M. Length-weight relationships of the tropical cephalopod Uroteuthis chinensis (Gray, 1849) from Sabah, Malaysia. Zoology and Ecology, 2014, 24 (3), 215-218.

Point 6: Comments on the Quality of English Language

English language is sufficient, some minor improvements are desirable.

Response 6: Thank for this valuable suggestion. The language has been improved by a professional organization.
